# AgentKit: Structured LLM Reasoning with Dynamic Graphs

**Yue Wu**[1,3]✉**, Yewen Fan**[1]**, So Yeon Min**[1]**, Shrimai Prabhumoye**[2,4]**, Stephen McAleer**[1]**,
Yonatan Bisk**[1]**, Ruslan Salakhutdinov**[1]**, Yuanzhi Li**[1,3]**, Tom Mitchell**[1]
[1]Carnegie Mellon University, [2]NVIDIA, [3]Microsoft, [4]Boston University
✉ ywu5@andrew.cmu.edu

## Abstract

We propose an intuitive LLM prompting framework (AgentKit) for multifunctional agents. AgentKit offers a unified framework for explicitly constructing a complex "thought process" from simple natural language prompts. The basic building block in AgentKit is a **node**, containing a natural language prompt for a specific sub-task. The user then puts together chains of nodes, like stacking LEGO pieces. The chains of nodes can be designed to explicitly enforce a naturally **structured** "thought process". For example, for the task of writing a paper, one may start with the thought process of 1) identify a core message, 2) identify prior research gaps, etc. The nodes in AgentKit can be designed and combined in different ways to implement multiple advanced capabilities, including on-the-fly hierarchical planning, reflection, and learning from interactions. In addition, due to the modular nature and the intuitive design to simulate an explicit human thought process, a basic agent could be implemented as simple as a list of prompts for subtasks and therefore could be designed and tuned by someone *without any programming experience*. Quantitatively, we show that agents designed with AgentKit achieve SOTA performance in WebShop and Crafter. These advances underscore the potential of AgentKit in creating effective and accessible agents for a wider range of applications. Github

## 1 Introduction

Recently, large language models (LLM) (Chowdhery et al., 2022; OpenAI, 2023; Manyika; Touvron et al., 2023; Jiang et al., 2023; Li et al., 2023; Parmar et al., 2024) have demonstrated remarkable performance in a variety of tasks, including embodied planning and acting (Wu et al., 2024a; Ahn et al., 2022; Du et al., 2023; Wang et al., 2023c; Wu et al., 2023a;b; Wang et al., 2023a), QA or dialogue (Ouyang et al., 2022; Hendrycks et al., 2020; Bubeck et al., 2023; Madaan et al., 2023), and general problem solving (Brown et al., 2020; Yao et al., 2022b; Shinn et al., 2023). However, two challenges remain to be overcome to apply LLMs to general real-world agent (Wooldridge & Jennings, 1995; Laird et al., 1987; Russell, 2010) tasks.

The first challenge is adhering to **procedural** requirements. For example, a self-driving car must adhere strictly to safety rules while making situational adaptations. The existing agent frameworks (Significant-Gravitas; Chase, 2022; Khattab et al., 2023; Shinn et al., 2023) do not follow explicit reasoning procedures. An error in one step can propogate and influence later steps (Yao et al., 2022b). Wu et al. (2024b) observes that simple chain-of-thoughts agents (Wei et al., 2022) generate good actions in the first few steps of the Tower of Hanoi, but their performance quickly degrades in later steps. Code generation (Wang et al., 2023c;a;b) attempts to address the first challenge by converting the procedural requirements into code.

However, code-based agents present the second challenge: **accessibility and ease of use**. Code-based agents rely on hand-crafted API platforms (Wang et al., 2023a) specialized for the task and often require many code examples that may be hard to produce. On the other hand, humans can make use of or transfer procedural knowledge effectively by tracing a series of thoughts or instructions in natural language.

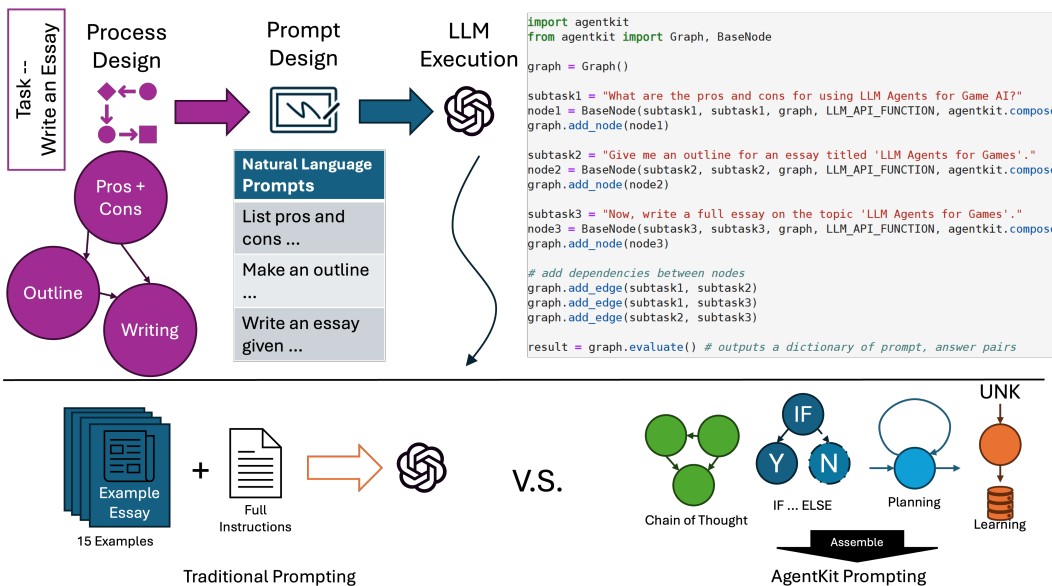

Figure 1: A user breaks down a task into subtasks (nodes) representing a "thought process" and creates prompts for the subtasks (nodes). Subtasks (nodes) in AgentKit can be designed and assembled in different ways to achieve diverse functionalities, similar to LEGO pieces.

To close the gap, we introduce AgentKit, an LLM prompting framework designed for *assembling* simple natural language subtasks into a solution to a complicated task. This design intuitively instructs an LLM to follow the predefined "thought process" by accomplishing all the subtasks before arriving at the final solution.

An agent in AgentKit consists of standardized units called *nodes*. Each node is designed to complete a specific subtask by sequentially performing 1) preprocess input (aggregate outputs from dependency nodes and an external database) and prompt the LLM, 2) postprocess the result for storage and use. For example, to operate a self-driving vehicle, an agent may be designed to first explicitly list and predict the intentions of entities such as pedestrians or other vehicles [node1], before deciding which driving action to take [node3]. Following a customized "thought process", AgentKit provides the user with precise modular control over the overall problem solving process, *without writing a single line of code*. This design also comes with the benefit of *modular interpretability*, allowing one to pin-point the error-causing subtask by examining the natural language outputs of each node.

To expand the capabilities, nodes and dependencies can be dynamically added and removed at inference time through coding, enabling complicated routing like IF...ELSE branching or FOR...LOOPS. For example, when road condition is bad, a self-driving agent can add a subtask to identify slippery roads [node2], before deciding the driving action [node3].

The set of dynamic nodes and dependencies naturally form a dynamic Directed A-cyclic Graph (DAG), with prompts as nodes and dependencies as edges. In our agent setting, we traverse the DAG computing the LLM results for each node in the topological order of the dynamic graph. We design one of the later nodes in the graph to directly output an environment action.

To demonstrate the full potential of AgentKit, we implement an agent for the Crafter (Hafner, 2021) game with several advanced capabilities: planning and dynamic goal prioritization; mistake identification and reflection; learning from experience. The agent consistently achieves SOTA performance and learns a good knowledge base from the environment. To show the generalization of AgentKit, we port a cost-effective version of our Crafter prompt to a web agent task and surpass SOTA by 5% on WebShop (Yao et al., 2022a).

Our key contribution is a framework for natural language "coding" of end-to-end multi-functional AI agents, with the ability to plan, reflect, and learn in challenging situations.

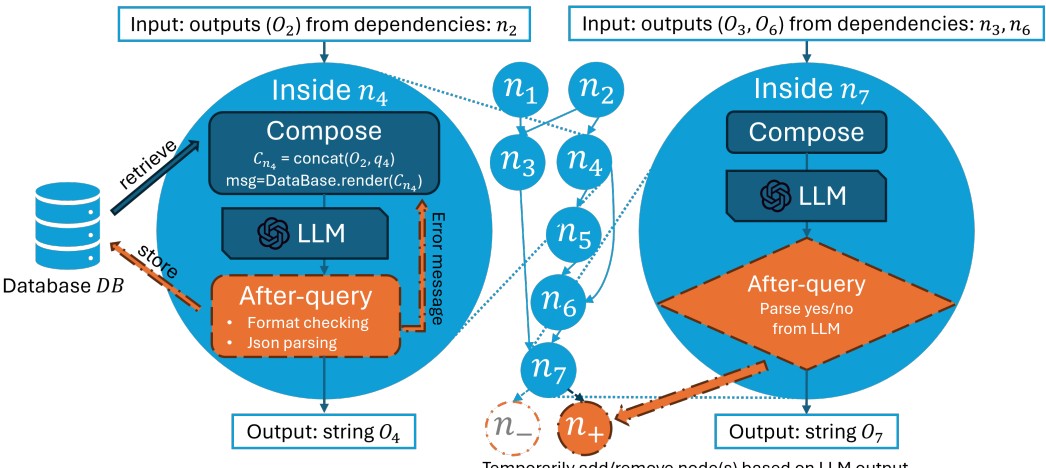

Figure 2: Each node in AgentKit takes outputs from its dependencies and outputs a string to complete a predefined subtask. The orange components (After-query) are optional and can be further customized with minimal programming through the AgentKit API. **Left:** The evaluation process inside a node consists of compose and after-query. **Right:** Nodes can be dynamically added / removed during the inference time. For example, the after-query operation of $n_7$ adds a conditional node $n_+/n_-$ based on a yes/no answer from the LLM to the node query. This induces conditional branching.

## 2 AgentKit

To use AgentKit, the user first defines a "thought process" as a set of natural language subtasks (prompts) that depend on each other. **Each subtask is presented to AgentKit as a node, centered around a natural language prompt for the subtask. The nodes are linked together by the dependency specifications in a directed acyclic graph (DAG)**, to implement different different logic and throught processes. Different arrangements of nodes could represent different functionalities, allowing the user to integrate various functionalities to build a multifunctional agent (Figure 1 (b)).

All nodes have access to a database $DB$. With a central database, the user can pass task specifications, instructions, and current game observation to every node in the graph. The database also allows nodes to store and pass on permananet information (e.g. plans or knowledge base) to future steps.

In this section, we explain the implementation details behind AgentKit and how it could be used, with a running example of a self-driving car. See the benefits of our codebase in C.

### 2.1 Node Components

For each node $v$, the user defines a subtask by specifying the prompt $q_v$ and a list of dependencies $D$. Inside the node, AgentKit runs a built-in flow that preprocesses the input (Compose), queryies the LLM with a preprocessed input and prompt $q_v$, and optionally postprocesses the output of the LLM (After-query). For example, node $n_4$ can be designed to *"Identify the intentions of other road users"* (left of Fig. 2).

#### 2.1.1 Compose: Gathering, compiling, and formatting information into the prompt

The compose operation takes the outputs of all dependencies and, optionally, information from the database to make a single prompt for the LLM. AgentKit comes with a default compose operation that can be used **without programming**. With the output from $v$ as $O_v$,

$$C_v = \texttt{concat}\left(DB[\text{``\{\$db.QUERY\$\}''}], \{O_d | d \in D\}, q_v\right) \quad (1)$$

For example, the compose operation for $n_4$ would gather the output of $n_2$ — "List any road-users in your surrounding." (Figure 2).

The compose operation could be further customized by coding a "compose function". This enables support for special system prompts and even RAG (Lewis et al., 2020).

### 2.1.2 Query/After-query: Parsing, formatting, modifying based on LLM outputs

The Query operation prompts the LLM with $C_v$ to get $LLM(C_v)$, and an *optional* after-query operation parses or re-formats $LLM(C_v)$, and edits the database $DB$. For example, $LLM(C_4)$, *"intentions of other road users"*, may need to be post-processed into Json format and added to the database, as illustrated in Algorithm 1 and Figure 2.

---
**Algorithm 1** After-query operation example

---
$A_v \leftarrow \mathcal{M}_{LLM}(C_v)$               ▷ initialize $A_v$ with the LLM response
result $\leftarrow$ Parse($A_v$)
**if** result does not match format **then**
     raise AfterQueryError("Message_for_LLM")    ▷ AgentKit catches the error and reties
**end if**
Update database $DB$ if needed
$O_v \leftarrow$ result                       ▷ set output to parsed result

---

## 2.2 Dynamic Components

To support advanced capabilities such as branching, AgentKit offers API for user to dynamically modify the DAG at inference time (Figure 2 right). All dynamic modifications are **temporary** and will be reverted at the end of a graph traversal pass. Note that modifications to nodes already evaluated in this pass are forbidden and will be automatically rejected. In the self-driving example, after identifying the intentions of road users, $n_7$ could be the task of "Identify if any road user poses the risk of collision". If the LLM output is "Yes", after-query of $n_7$ could add a node $n_+$ that *"adjust route in order to avoid the road users"*.

**Temporarily removing edges/nodes.** A typical use case can be to conditionally skip a set of unnecessary nodes to save computation if the node results can be reused in this pass. For example, in section 3, if the plan does not require update, the planner questions are skipped.

**Adding temporary nodes/edges.** A typical use case is to create conditional branches. For example, a node could direct the flow by adding conditional edges and nodes based on a "yes/no" answer from the LLM.

## 2.3 Dynamic Graph Traversal

We implement Kahn's Algorithm (Algorithm 2) to traverse the DAG of both static nodes and dynamically added/ removed nodes. Since there may exist more than one topological order for a given graph, the dynamic addition/removal of components may result in non-deterministic or unexpected behaviors. Safeguards are put in place to catch potential unexpected behaviors (section A).

# 3 Example Agent Powered by AgentKit

We demonstrate an agent for the Crafter (Hafner, 2021) game with several advanced capabilities: hierarchical planning, reflection, and learning from interactions. At each step $T$, the agent takes the text description ($o_T$) of the visual game observation and an instruction manual $\mathcal{I}$ (Appendix D) as input similar to Wu et al. (2024b), and outputs an action index ($a_T$) the number of times the action should be repeated ($n_T$). All features are implemented with prompts on a single graph in AgentKit (complete list of prompts in Appendix F.1).

As illustrated in Figure 3 (a), at every step, the agent summarizes and keeps track of: 1) observation ($O^T_{n_{\text{s-obs}}}$); 2) plan and reasoning ($O^T_{n_{\text{s-plan}}}$); 3) infer if the previous action succeeded based on changes in the observation ($O^{T-1}_{n_{\text{s-action}}}$).

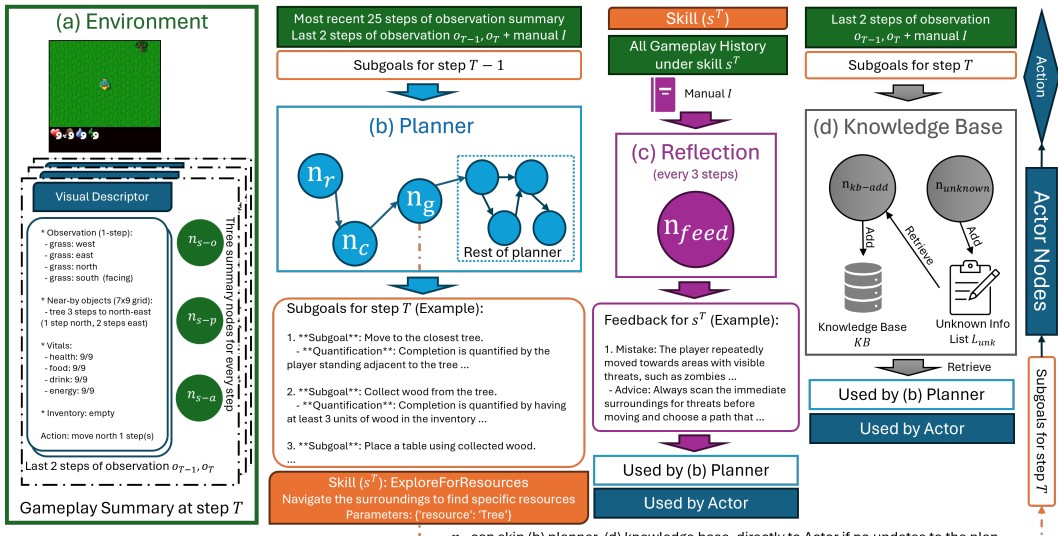

Figure 3: Node names are abbreviated for space. **(a)** At every step in the game, three summary nodes (green) $n_{\text{s-obs}}$, $n_{\text{s-plan}}$, $n_{\text{s-action}}$ summarize the observation, plan, and action of the current step. **(b)** At step $T$, all planner nodes (blue) take $o_{T-1}$, $o_T$ and manual $\mathcal{I}$ as input, and output 3 subgoals and a skill $s^T$. $n_{\text{reflect}}$ reflects on the summary of the 25 most recent steps, and $n_{\text{challenge}}$, $n_{\text{gate}}$ determines whether the subgoals for $(T-1)$ are carried over or updated. **(c)** Every 3 steps under skill $s^T$, ($n_{\text{feed}}$ purple) reflects on all gameplay history under $s^T$ and generates a skill specific feedback for the planner (b). **(d)** Every step $T$, $n_{\text{kb-add}}$ (gray) examines $o_{T-1}$, $o_T$ and $\mathcal{I}$ to identify new information from $L_{unk}$. $n_{\text{unknown}}$ adds to $L_{unk}$ by identifying missing information from $\mathcal{I}$ for the current sub-goal.

## 3.1 Hierarchical Planner with Short-term Reflection (Figure 3 b)

At step $T$, the LLM first generates *a summary and reflection* ($n_{\text{reflect}}$) of the previous 25 steps:

$$\mathcal{M}_{LLM}\left(concat\left(\left\{concat(O^t_{n_{\text{s-obs}}}, O^t_{n_{\text{s-plan}}}, O^t_{n_{\text{s-action}}})|t \in (T-25, T]\right\}\right), q_{\text{reflect}}\right) \quad (2)$$

The LLM is then prompted to *identify the top challenge of 3 of the most important high-level challenges* (node $n_{\text{challenge}}$). Then node $n_{\text{gate}}$ *determines if the sub-goals from the previous step $(T-1)$ need to be updated based on $O_{n_{\text{reflect}}}$, $O_{n_{\text{challenge}}}$ and the "completion criteria"*.

If an update is not required, the planner and knowledge base are skipped directly to the Actor nodes, saving more than 1/3 the number of tokens. Otherwise, node $n_{\text{subgoal}}$ prompts the LLM to *identify 3 subgoals in the order of execution, each paired with a "completion criteria"*.

Using information on the top sub-goal, the agent keeps track of a list of "skills" ($n_{\text{skills}}$) that prompts the LLM to *retrieve and create new skills to build a skill library $\mathcal{S}$* similar to Wang et al. (2023a), implemented as a Python dictionary containing LLM generated attributes like: skill_name, skill_description, etc. See Figure 3 (b) and Appendix E.2 for details.

## 3.2 Long Term Reflection (Figure 3 c)

Every 3 steps under a skill $s \in \mathcal{S}$, we compute a skill-specific feedback by prompting the LLM with all previous steps under skill $s$.

$$O^T_{n_{\text{feed}}} = \mathcal{M}_{LLM}\left(concat\left(\left\{concat(O^t_{n_{\text{s-obs}}}, O^t_{n_{\text{s-plan}}}, O^t_{n_{\text{s-action}}})|s^t = s \wedge t \leq T\right\}\right), q_{\text{feed}}\right) \quad (3)$$

The feedback $O_{n_{\text{feed}}}$ is provided as context to both the actor and the planner.

## 3.3 Learning from Interaction with a Knowledge Base (Figure 3 d)

The agent uses two carefully designed prompts (nodes): $n_{\text{unknown}}$, $n_{\text{kb-add}}$, to maintain an unknown information list $L_{\text{unk}}$ and a knowledge base $KB$.

| Method | Score | Reward | Costs[1,3] |
|---|---|---|---|
| Human Experts | $50.5 \pm 6.8\%$ | $14.3 \pm 2.3$ | N/A |
| AgentKit (Model mix*) | $20.64\%$[‡] | $\mathbf{12.8 \pm 2.1}$ | \$138 |
| SPRING (GPT-4) (Wu et al., 2023b) | $\mathbf{27.3 \pm 1.2}\%$ | $12.3 \pm 0.7$ | \$243[†] |
| DreamerV3 (Hafner et al., 2023) | $14.5 \pm 1.6\%$ | $\mathbf{11.7 \pm 1.9}$ | 1M steps |
| EDE (Jiang et al., 2022) | $11.7 \pm 1.0\%$ | N/A | 1M steps |
| DreamerV2 (Hafner et al., 2020) | $10.0 \pm 1.2\%$ | $9.0 \pm 1.7$ | 1M steps |
| SPRING (GPT-4-turbo on official repo) | $7.74\%$[‡] | $7.1 \pm 3.4$ | \$85 |
| ELLM (Du et al., 2023) | N/A | $6.0 \pm 0.4$ | 5M steps |
| Rainbow (Hessel et al., 2018) | $4.3 \pm 0.2\%$ | $5.0 \pm 1.3$ | 1M steps |
| PPO (Schulman et al., 2017) | $4.6 \pm 0.3\%$ | $4.2 \pm 1.2$ | 1M steps |
| CoT (Wei et al., 2022) (GPT-4-turbo) | N/A | $3.9 \pm 1.6$ | \$20 |
| Plan2Explore (Sekar et al., 2020) | $2.1 \pm 0.1\%$ | $2.1 \pm 1.5$ | 1M steps |
| RND (Burda et al., 2018) | $2.0 \pm 0.1\%$ | $0.7 \pm 1.3$ | 1M steps |
| Random | $1.6 \pm 0.0\%$ | $2.1 \pm 1.3$ | 0 |

Table 1: Table comparing AgentKit and popular RL algorithms in terms of reward, and training steps. The results for AgentKit are summarized over 3 independent trials. *Our AgentKit implementation uses a combination of GPT-4-0613, GPT-4-turbo, and GPT-3.5-turbo to balance cost and performance. [‡] Score is computed on a group of 3, we did not run multiple groups for std. [†]Cost of GPT-4 estimated from token counts of GPT-4-turbo.

At each step $T$, $n_{kb\text{-}add}$ prompts the LLM to *identify the information from $L_{unk}$ by reasoning with the last two steps of observation $o_{T-1}, o_T$*. Indentified information is stored in the knowledge base $KB$. Then, $n_{unknown}$ prompts the LLM to *list missing information (from the instruction manual and knowledge base) required for the current sub-goal*. The list of missing information is then added to $L_{unk}$. We show a learned KB in Appendix E.3

## 4 Experimental Results

We present our experiments as follows. First, we use Crafter (Hafner, 2021) for benchmarking and analysis of different modules, described in section 3. We then describe our experimental results for WebShop (Yao et al., 2022a) to demonstrate the generalization of AgentKit.

### 4.1 Crafter

Crafter (Hafner, 2021) is an open-world survival game featuring procedural generation, designed to benchmark RL algorithms. The game includes a tech tree with 22 achievements that span 7 levels. The original observation of the game is a top-down view of the shape $(7 \times 9)$, with a discrete action space comprising 17 options. Drawing inspiration from Minecraft, the game presents similar challenges including crafting, exploration, survival, and progressing the tech-tree. Following Du et al. (2023); Wu et al. (2023b; 2024b), we provide the text description and instructions from Wu et al. (2023b) as input to the agent. The text description includes: 4 blocks to the agent's immediate surrounding, the closest 2 objects of each kind within view, the agent vitals, and a list of remaining achievements. We show an example description in Appendix E.1. Additionally, we allow the agent to output a *repeat_number* to repeat the chosen action without requerying the agent. This feature reduces the cost by 1/3 on average.

Notably, the LATEX source code (Hafner, 2021), from which Wu et al. (2023b) extrated the instructions, omitted many crucial details about the actual game, since the research paper was never intended to be a game manual. Omitted details include: specific amount of materials required to craft table (2 woods) and furnance (4 stones), damage/attack range of weapons or enemies. Such an omission poses a great challenge to first-time human players, making the game essentially unsolvable without trying and learning from the environment.

### 4.1.1 Crafter Results

We compare the performance of AgentKit with SPRING (Wu et al., 2023b) and RL baselines[1]. Note that the original version of SPRING uses GPT-4, which is expensive to run on Crafter. The GPT-4-turbo is significantly cheaper and features improved reproducibility and instruction following[2]. Although consistency and instruction following is generally considered a desirable feature, it increases the risk of getting stuck, for example, in situations when the instruction manual is inaccurate. Also, we observe that GPT-4-turbo is worse at spatial reasoning compared to GPT-4. We hypothesize that the two differences explained above are the cause of the deteriorated performance with GPT-4-turbo (Table 1 row 7).

In contrast to SPRING, AgentKit features active reflection and learning capabilities, along with explicit planning. Therefore, our agent does not rely on randomness to get past unexpected situations (section 4.1.2). Overall, our agent achieves 80% improvement in terms of reward compared to SPRING (GPT-4-turbo) and attains reward comparable to SPRING (GPT-4) while being 45% cheaper[3]. The consistency of our agent over SPRING is also reflected in the score metric, which favors low unlock rates for more achievements over consistently high unlock rates for less achievements (Hafner, 2021). Empirically, we observe that SPRING tend to randomly unlock less important achievements i.e. Eat Plant, Collect Sampling, etc. Such behavior improves the score metric but does not affect the reward.

### 4.1.2 Planning, Reflection, and Learning from Interactions

We show an illustrated example of the first 11 steps taken by our Crafter agent in Figure 4.1.1. The agent started empty handed in the environment (step 0) and decided to approach the challenge $n_c^0$ of "Resource Collection". Given the challenge, it generated the sub-goal of "Move toward tree". After reaching the tree at step 6, the short-term reflection node $n_r^6$ and the planer node $n_g^6$ detects the completion of the sub-goal, and updated the plan and action to collect wood and place table. Simultaneously, "TableWoodConsumption" was identified as missing from the manual and added to the unknown information list ($L_{unk}$) by $n_{\text{unknown}}$.

However, due to the omitted information (Hafner, 2021) on the amount of wood required for the table, the agent attempts to place the table with 1 wood and fails (step 8). The observation nodes ($o_{s-action}^8$) automatically detect the failure by comparing the current observation with the expected action outcomes. The observation nodes then referred to $L_{unk}$ and the manual $\mathcal{I}$ and identified "Insufficient wood" to be the cause of failure. The plan and actions are updated accordingly to gather more wood before placing the table. Finally, "TableWoodConsumption" is discovered as 2 and added to the knowledge base $KB$ (step 11).

Finally, the long-term reflection node $n_{\text{feed}}$ identifies failed movement attempts due to obstacles and insufficient resource collection. The planner is suggested to "ensure flat ground before movement" and "prioritize wood collection".

## 4.2 Webshop

Creating autonomous agents for the Web could bring convenience to our lives. We demonstrate a zero-shot agent for Webshop (Yao et al., 2022a), a simulated virtual shopping environment. A shopping task in Webshop involves searching, browsing, and identifying the desired products on a simulated e-Commerce platform. Due to the specific nature of the shopping task, all prior LLM agents rely on few-shot demonstrations of complete human trajectories (Yao et al., 2022b; Shinn et al., 2023; Liu et al., 2023) to guide agent behavior. Since AgentKit can customize agents with nodes, our agent does not require demonstrations.

Inspired by the agent design for Crafter (section 3), we design a simple 6-node agent with short-term reflection and planning similar to the Crafter agent. We include the complete

---

[1]The 1M cap for number of interactions is set by the benchmark developers (Hafner, 2021)
[2]platform.openai.com/docs/models/gpt-4-and-gpt-4-turbo
[3]Cost calculated based on official API pricing as of Mar 2024: openai.com/pricing.

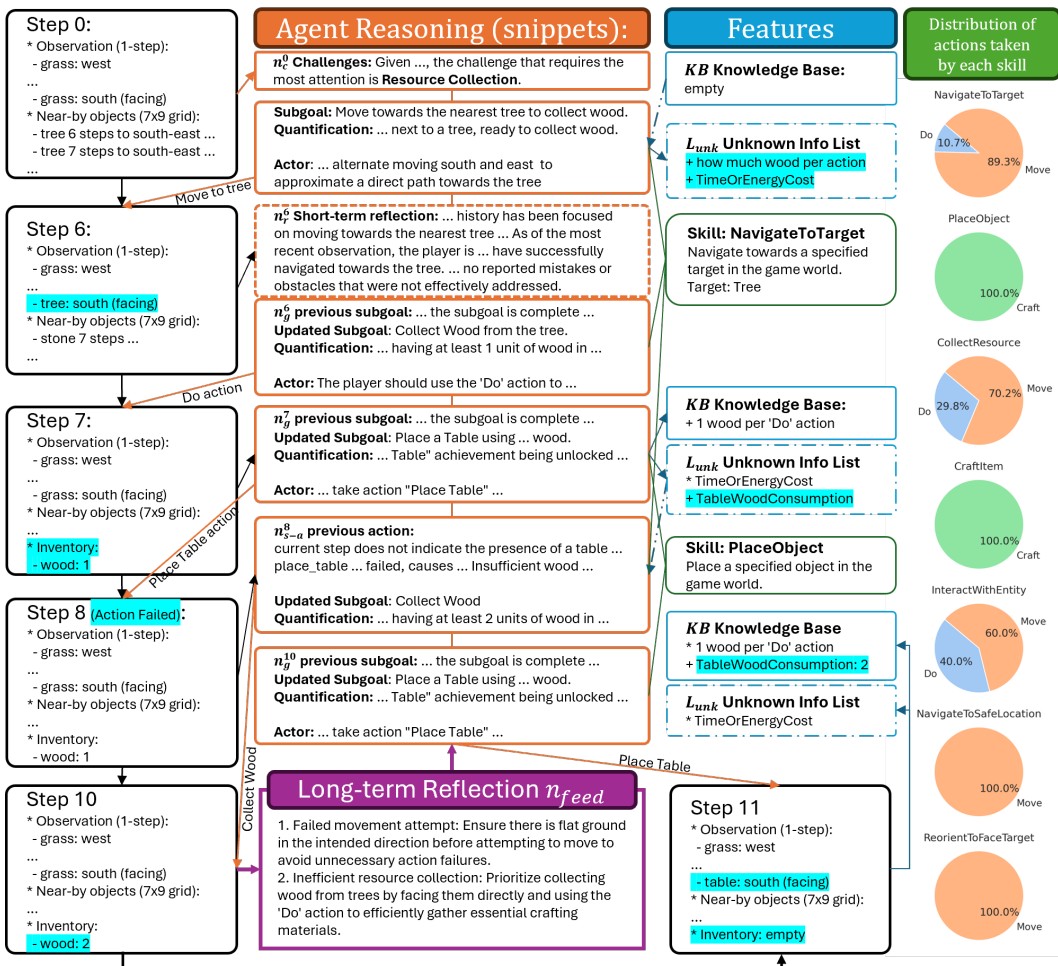

Figure 4: **Left three columns:** an example trajectory in Crafter. Different nodes on planning, reflection, feedback, knolwedge discovery work together to complete the first 11 steps and successfully crafting the table. Through environment interactions and error identification/-correction, the agent discovered two pieces of information regarding "wood per Do action" and "table wood consumption", originally omitted by the instructions (Hafner, 2021). **Right column:** the end-of-game distribution of all actions (classified into categories of Move, Do — Interact, Craft) taken by the agent, for each skill in the skill library. The action distribution aligns well with human expectations based on skill names.

prompts in Appendix F.2. As shown in Table 2, our zero-shot agent achieves a 5% performance improvement compared to SOTA few-shot agent, ReAct (Yao et al., 2022b).

| Method | Act (Yao et al., 2022b) PaLM-540B | ReAct (Yao et al., 2022b) PaLM-540B | AgentBench (Liu et al., 2023) Best of GPT-4/3.5 | AgentKit GPT-4-turbo | AgentKit GPT-4-0631 |
|---|---|---|---|---|---|
| Score | 62.3% | 66.6% | 64.1% | **69.4**% | **70.2**% |

Table 2: Table comparing our agent against baselines on WebShop (Yao et al., 2022a). For the sake of cost, we report the score on the first 100 samples of WebShop. Our agent achieves SOTA performance using both GPT-4 and the lower-cost GPT-4-turbo.

# 5 Related Works

## 5.1 LLM for Planning

Early attempts to use LLM for agent tasks often involve high-level planning. Huang et al. (2022); Ahn et al. (2022) proposed generating abstract high-level plans for embodied tasks,

with limited action space and trajectory length. Song et al. (2022); Wu et al. (2023a) improves Ahn et al. (2022), introducing more complexity in the task and the planner. Du et al. (2023); Wang et al. (2023c); Yuan et al. (2023) adopts similar LLM planner framework in the domain of open-world games like Crafter and Minecraft. Prior works along this line all share the same prompt design of providing expert or human trajectory examples as context for the LLM. The output of the LLM is used as a static high-level plan, independent of the agent's situation.

## 5.2 LLM Agents

In a more interactive setting, the LLM typically has access to some status information of the agent and can react to environment signals. Yao et al. (2022b) solves simple natural language tasks through LLM interaction with chain-of-thought prompts. Shinn et al. (2023) further adds the ability to read task feedback signals. Wang et al. (2023a) plays Minecraft with an LLM that creates, retrieves, and refines code from a skill library. Wang et al. (2023b) uses a VLM for high-level planning and re-planning in Minecraft. However, the actual generated plans are often less than 10 stpes. In addition, the LLM operates independent from the rest of the agent (i.e., low-level controllers execute the plan and the LLM planner receives a success/fail message) and cannot actively react to the environment signals. This design limits the agent in Minecraft to peace_mode (Wang et al., 2023b) due to the inability to react to survival situations, an essential part of Crafter (Hafner, 2021).

Notably, all above works rely on a few-shot prompting. Although few-shot examples are useful for enforcing specific output formats, they tend to become long and unhelpful for complex tasks. Even writing high-level householding requires up to 17 full demonstrations (Ahn et al., 2022), making it costly for long-horizon tasks such as Crafter (Wu et al., 2023b).

## 5.3 Structured prompting for LLMs

Wei et al. (2022) first showed the benefit of explicit reasoning through chain-of-thought prompts. Yao et al. (2023); Besta et al. (2023) demonstrated success in guiding traditional search algorithms with LLM scoring functions, but the applications are limited to search-based problems. Khattab et al. (2023) parameterize prompts with trainable NN layers, optimized for specific metrics. Chase (2022); Beurer-Kellner et al. (2023) offers pre-built implementations of different prompting frameworks under a unified code base, but introduces a lot of coding hasle. Unlike existing works that identify a particular structure of structured prompting (Yao et al., 2022b; Shinn et al., 2023; Madaan et al., 2023) (e.g., the sequence of generate, self-evaluate, self-reflect), we present a modular perspective (Andreas et al., 2016) of designing and building agents from simple subtasks, with mostly intuitive natural language specification.

The most similar to AgentKit is Wu et al. (2023b), which uses LLM reasoning by traversing a fixed DAG to achieve SOTA performance in Crafter. Our framework adds dynamic components, pre-/post-processing, and a database on top of Wu et al. (2023b). The additional capabilities allow us to implement additional features like hierarchical planning, reflection, and knowledge base.

## 6 Conclusion

In this work, we present a framework (AgentKit) that allows a user to intuitively create a specific "thought process" for a complex problem by assembling prompts for subtasks. AgentKit evaluates LLM for each prompt in the order defined by the dependencies to achieve consistent and structured reasoning. While the basic features in AgentKit can be used without coding experience (Figure G), custom Python preprocess and postprocess functions can enable advanced features such as database, conditional branching, and output formatting / parsing. To demonstrate the power and versatility of AgentKit, we implement a WebShop (Yao et al., 2022a) agent and a Crafter (Hafner, 2021) agent, both achieving

SOTA performance. We further demonstrate advanced capabilities that include on-the-fly hierarchical planning, reflection, and learning from interactions in the Crafter agent.

Our work demonstrates an intuitive way to prompt an LLM for challenging tasks by explicitly writing a human thought process into the prompts. Given the strong results, we envision AgentKit as an early demonstration of "coding" end-to-end multifunctional AI agents to solve challenging tasks, even for humans, with natural language.

## 7 Limitations

Two main limitations remain for AgentKit: Increased LLM querying **cost** and **manual efforts** for process design and prompt creation. Majumder et al. (2023); Nottingham et al. (2024) propose to learn prompts from environment rewards. However, experiments are conducted with limited task length ($<= 5$ steps), and scaling the learning to long-horizon tasks featuring environment noise and concurrent objectives remains an open problem.

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

# Appendix

## Table of Contents

## A Safeguards for Dynamic Operations

For dynamic add/remove of edge $(u, v)$, AgentKit asserts: $v$ have not been evaluated for add and both $u, v$ have not been evaluated for remove.

The assert statements avoid not only loops, but also unpredictable behaviors. For example, a graph $A - F, C - F$ has multiple topological orders. A dynamic edge $C - A$ could potentially alter the previous perceived topological order. The assert statements raise it to the attention of the user. A loop can still be implemented by adding new nodes (chain $A - B - \cdots$ that keeps growing) and it is up to the user to prevent infinite loops.

## B Kahn's Algorithm

---

**Algorithm 2** Kahn's Algorithm (Kahn, 1962) for topological transverse

---

1: $L \leftarrow$ empty list that will contain the sorted elements
2: $F \leftarrow$ set of all nodes with no incoming edges         ▷ Initialize the frontier $F$
3: $inDegree \leftarrow$ map of node to in-degree
4: **while** $F$ is not empty **do**
5:     remove a node $n$ from $F$         ▷ Retrieve the next node from the frontier $F$
6:     add $n$ to the end of $L$ and evaluate $n$         ▷ Evaluate the node
7:     **for all** node $m$ with an edge $e$ from $n$ to $m$ **do**
8:         $inDegree[m] \leftarrow inDegree[m] - 1$
9:         **if** $inDegree[m] = 0$ **then**
10:            insert $m$ into $F$       ▷ Add the next node with in-degree 0 to the frontier $F$
11:         **end if**
12:     **end for**
13: **end while**
14: **return** $L$         ▷ Topological traversal successful

---

## C Design

Our implementation in graph.py provides a solid foundation for managing complex graph structures, emphasizing support for both permanent and temporary nodes and edges, in addition to dynamic evaluation and modification capabilities. This design is firmly rooted in Object-Oriented Programming (OOP) principles, ensuring that the AgentKit API remains flexible, extendable, and easy to maintain.

### C.1 Encapsulation

The Graph class effectively encapsulates the complex logic required for managing graph elements, including nodes and edges (whether permanent or temporary), and determining

the execution order of these elements. By hiding the internal state and offering a carefully selected suite of methods for interacting with the graph, we ensure that the class remains accessible and secure against unintended manipulations. Inheritance and Polymorphism Our design philosophy accommodates extensibility through inheritance, enabling developers to derive specialized graph types from the Graph class.

### C.2 Extensibility

The Graph class was conceived with future growth in mind. Methods such as add_node, add_temporary_node, add_edge, and add_edge_temporary empower developers to adapt the graph structure dynamically, reflecting real-time changes or experimental scenarios. This distinction between temporary and permanent graph components allows one to keep clear track of the original structure, while maintaining a level of flexibility that is critical for advanced logics like branching.

Moreover, the architecture supports enhancements through subclassing, opening avenues for introducing novel node or edge types with unique characteristics or for deeper integration with external frameworks, such as wandb_LLM for advanced logging and traceability.

## D Manual

### D.1 Crafter

```
List of desired interactions:
 – avoid zombies, skeletons, and spiders.
 – collect saplings.
 – craft a wood pickaxe.
 – collect wood.
 – craft a stone pickaxe.
 – collect stone.
 – craft a furnace.
 – collect coal.
 – collect iron.
 – craft an iron pickaxe.
 – collect diamonds.
 – craft an iron sword.
 – chase cows.
 – grow fruits.
 – drink from a lake.
 – sleep in a safe place.
 – craft a table.
 – eat food.
 – drink water.
 – rest.
 – build stone tools to defend myself against monsters.
 – build bridges to cross lakes.
 – dig tunnels to hide from monsters.
 – block arrows with stones.
 – dig through walls to surprise skeletons.
 – seek shelter in caves.
 – build plantations of saplings and defend them against monsters.
 – eat the growing fruits to ensure a steady food supply.
 – place a table.
 – eat a cow.
 – place a plant.
 – defeat a zombie.
 – place stone.
 – eat a plant.
 – defeat a skeleton.
 – wake up.
 – place a furnace.
```

List of game achievements and their requirements:
1. Collect Wood: No requirements
2. Place Table: Requires Collect Wood
3. Eat Cow: No requirements
4. Collect Sampling: No requirements
5. Collect Drink: No requirements
6. Make Wood Pickaxe: Requires Place Table
7. Make Wood Sword: Requires Place Table
8. Place Plant: Requires Collect Sampling
9. Defeat Zombie: No requirements
10. Collect Stone: Requires Make Wood Pickaxe
11. Place Stone: Requires Collect Stone
12. Eat Plant: Requires Place Plant
13. Defeat Skeleton: No requirements
14. Make Stone Pickaxe: Requires Collect Stone
15. Make Stone Sword: Requires Collect Stone
16. Wake Up: No requirements
17. Place Furnace: Requires Collect Stone
18. Collect Coal: Requires Make Wood Pickaxe
19. Collect Iron: Requires Make Stone Pickaxe
20. Make Iron Pickaxe: Requires Place Furnace, Collect Coal, and Collect Iron
21. Make Iron Sword: Requires Place Furnace, Collect Coal, and Collect Iron
22. Collect Diamond: Requires Make Iron Pickaxe

List of all actions and their requirements:
1. Move West: Flat ground to the west of the agent.
2. Move East: Flat ground to the east of the agent.
3. Move North: Flat ground to the north of the agent.
4. Move South: Flat ground to the south of the agent.
5. Do: Facing creature or material; have necessary tool.
6. Sleep: Energy level is below maximum.
7. Place Stone: Stone in inventory.
8. Place Table: Wood in inventory.
9. Place Furnace: Stone in inventory.
10. Place Plant: Sapling in inventory.
11. Make Wood Pickaxe: Nearby table; wood in inventory.
12. Make Stone Pickaxe: Nearby table; wood, stone in inventory.
13. Make Iron Pickaxe: Nearby table, furnace; wood, coal, iron an inventory.
14. Make Wood Sword: Nearby table; wood in inventory.
15. Make Stone Sword: Nearby table; wood, stone in inventory.
16. Make Iron Sword: Nearby table, furnace; wood, coal, iron in inventory.
17. Noop: Always applicable.

Health restores automatically over time, independent from food and hydration.

Notes:
 – Diagonal actions are not supported, only use the four cardinal directions.
 – The game world is infinitely large and procedurally generated from a fixed random seed.
 – If you are within close proximity to a zombie, it will chase you. You must kill the zombie to survive.
 – When sleeping, the player will not be able to take any actions until energy is full and will take triple damage from zombies. Therefore, do not sleep when threats are nearby.

### D.2 Webshop

 – Try to make a purchase in 25 steps.

– Puchase a product with no less than 80% match with the task. No need to
    look for a perfect match. If multiple products match the task,
    purchase the one with the highest match.
– The search engine uses bag-of-words. Optimize the search terms
    accordingly and don't put price in search. If you cannot find
    relevant items, consider going to the next page of the search results
    .
– The search result page only contain short product descriptions. Click
    on the product to reveal other customizable attributes/options.

## E    Examples

### E.1    Observation Description

== Gamestep 205 ==

```
* Observation (1-step):
  – zombie: west, 1W
  – zombie: east, 1E
  – grass: north, 1N (facing)
  – grass: south, 1S

* Near-by objects (7x9 grid):
  – tree 2 steps to west, 2W
  – tree 3 steps to north-west, 2N 1W
  – table 5 steps to south-east, 2S 3E
  – zombie 1 steps to west, 1W
  – zombie 1 steps to east, 1E

* Further to the north: 25 grass(s), 2 tree(s).

* Vitals:
  – health: 1/9
  – food: 7/9
  – drink: 6/9
  – energy: 9/9

* Inventory:
  – sapling: 1
  – coal: 1
  – iron: 1
  – wood_pickaxe: 1
  – stone_pickaxe: 1

Action:
move_south 1 step(s)
```

### E.2    Skill Library generated by the Crafter agent

```
{
"NavigateToLocation": "Move towards a specific location using cardinal directions.",
"CollectResource": "Interact with an environment object to collect a resource.",
"PlaceObject": "Place an object in the game world from the player's inventory.",
"CraftItem": "Craft a specific item using available resources and tools.",
"UseFurnace": "Use a furnace to smelt or craft items.",
"InteractWithEntity": "Interact with an entity in the game world to achieve a specific outcome.",
"Rest": "Restore energy by sleeping or resting in a safe place."
}
```

### E.3    Knowledge Base learned by the Crafter agent

We note a mistake in "Exact_Quantity_of_Stone_Required", the correct value should be 1.

```
{
  "Collecting_Wood_Requirements": "No specific tool needed to collect wood.",
  "Direct_Interaction_with_Trees": "Direct interaction with trees possible.",
  "Wood_Quantity_for_Table": "2 wood for table",
  "Wood_Collection_Amount_Per_Action": "1 wood per action",
  "Energy_Cost_for_Movement": "Movement does not consume energy.",
  "Crafting_Location_and_Proximity": "Adjacent to table for crafting.",
  "Exact_Quantity_of_Stone_Required": "3 units of stone for Stone Pickaxe.",
  "Exact_Quantity_of_Wood_Required": "1 unit of wood for Stone Pickaxe.",
  "Interaction_with_Environment_for_Wood_Collection": "'Do' action for wood collection.",
  "Quantity_of_Coal_Collected_Per_Action": "1 coal per action",
  "Specific_Action_for_Collecting_Coal": "'Do' action collects coal",
  "Use_of_Stone_Pickaxe_for_Coal": "Stone Pickaxe not required for coal",
  "Cues_for_Water_Sources": "Water sources indicated as 'water' in observations.",
  "Action_to_Collect_Drink": "'do' action collects drink",
  "Impact_of_Collecting_Drink_on_Status": "Increases drink level",
  "Proximity_to_Water_Source_for_Collection": "Adjacent to water for collection",
  "Quantity_of_Drink_Collected_Per_Action": "+1 drink per action",
  "Tool_Requirement_for_Collecting_Drink": "No tool required",
  "Crafting_Location_Requirement_for_Furnace": "No table required for furnace placement.",
  "Placement_Conditions_for_Furnace": "Flexible furnace placement.",
  "Tool_Requirements_for_Crafting_Furnace": "No tool required for furnace.",
  "Environmental_Obstacles": "Trees block movement.",
  "Cow_as_Obstacle_for_Movement": "Cows block movement",
  "Proximity_Requirements_for_Do_Action": "Must face and be near tree for 'Do' action.",
  "Wood_Sword_Crafting_Requirements": "Wood and table needed for wood sword.",
  "Crafting_Location_Specifics_for_Wood_Sword": "Near table, facing not specified.",
  "Crafting_Process_Details_for_Wood_Sword": "'Do' action near table with wood.",
  "Energy_Consumption_for_Movement": "Movement does not consume energy.",
  "Energy_Requirements_for_Movement": "Moving does not consume energy.",
  "Feedback_Mechanism": "Feedback through changes in observation, status, or inventory.",
  "Impact_of_Actions_on_Vitals": "Not all actions impact vitals directly.",
  "Creature_Movement_Blocking": "Creatures block movement.",
  "Feedback_Mechanisms_Post_Movement": "Observation changes post-movement.",
  "Interaction_with_Cow": "Adjacency required for cow interaction.",
  "Proximity_to_Targets": "Adjacency needed for interaction.",
  "Cow_Movement": "Cows can move closer.",
  "Immediate_Effects_of_Movement_on_Player_Status": "No immediate effect on vitals or inventory.",
  "Environmental_Changes_Post-Movement": "Environment relative to player changes with movement.",
  "Eating_Cow_Process": "Direct 'Do' action eats cow.",
  "Impact_on_Food_Level_from_Eating_Cow": "Food level increased significantly.",
  "Tool_Requirement_for_Eating_Cow": "No tool required."
}
```

# F  Prompts

## F.1  Crafter Prompts

Lots of prompts require custom pre/postprocessing. See github repo for full implementation.

```
{
'obs_obj':{
'prompt':"""
First, describe all objects the player faces or around the player. Describe the object type, the direction, the distance, the
↪  coordinates, and the requirements to interact with the object from the instruction manual.
Be precise and accurate with the direction, coordinates.
Output a Json list in the following format:
[
{"object":$type, "direction":$precise_direction, "distance":$distance$, "facing":$[yes/no], 'coordinate':"precise coordinates",
↪  "requirements":"requirements to interact from instruction manual, put 'NA' if not applicable"},
...
]

Second, in one sentence, describe what the player sees further to the front.
""",
'dep':[],
},
```

```
'obs_inv':{
'prompt':"""Describe the player's inventory as a json dictionary, with the item_names as key and item_amount as value. Write '{}'
↪  if inventory is empty.""",
'dep':[],
},

'obs_vit':{
'prompt':"""Describe the player's vitals as a json dictionary, in the format of {vital_name: "value/max"}.""",
'dep':[],
},

'obs_chg':{
'prompt':"""Using a list, document any changes to the player observation, inventory, and status in the most recent step.
Be precise about changes to numbers and details about the vitals, inventory, and the distance to objects around the player.
Be concise and only list the changes. Output 'NA' if there is no change""",
'dep':[],
},

'obs_last_act':{
'prompt':"""First, consider how the most recent action should have changed the player's observation/status/inventory, be specific
↪  with the action and the target.
Then, deduce if the last action succeeded.
If yes, end the answer here.
If not, identify potential cause for the failure, by focusing on the player's surrounding/observation, inventory, instruction
↪  manual, and missing information.
Be concise with the language, but precise with the details.""",
'dep':['obs_chg',],
},

's-obs-obs':{
'prompt':"""In one sentence, describe the observation and inventory. Include all observed object types and inventory items.""",
'dep':['obs_obj','obs_inv'],
},

's-obs-vitals':{
'prompt':"In one sentence, describe the current vitals.",
'dep':['obs_vit'],
},

's-action':{
'prompt':"""Output a Json dictionary of the following format:
```
{
"action": $action, # The most recent action, including the direction if it's a movement action
"repeats": $repeats # The number of times the action was repeated
"target": $target # The target of the action. Write 'NA' if the action is a movement/sleep action.
"success": $success # [yes/no] If the action succeeded
"causes_of_failure": $causes_of_failure # In one sentence, write the cause(s) of the failure. Write 'NA' if the action succeeded.
}
```
""",
'dep':['obs_last_act',],
},

'obs_current_actions':{
'prompt':"""For each action in the numbered list of all actions, reason with the current observation/surroundings and identify if
↪  the action is allowed at the current gamestep according to the requirements.
Then predict the target the action will interact with.
Format the output as a Json dictionary of the following format:
{
"$action": {"requirements": $requirements of the action, "related observation": $what in the observation may affect/block the
↪  action?, "reasoning": $precise but very concise reason of why the action is blocked, "allowed": $yes/no, "target": $inferred
↪  target, "unlock new achievement": $yes/no, will the success of this action unlock an unaccomplished achievement?}, # do not
↪  output white spaces
...
}
""",
'dep':['s-action', 'obs_obj'],
},

'reflect':{
'prompt':"""Consider how the player reached the current state, concisely summarize the player's high-level trajectory in the
↪  gameplay history. Focus on the present and the past and do not make future predictions.
Has the player been making effective progress towards the top subgoal '$db.subgoals.subgoal$'? Has the player been effectively
↪  addressing all the mistakes/obstacles encountered?
Output 'NA' if there is no history.""",
'dep':['obs_obj', 'obs_inv', 'obs_vit', 'obs_chg', 'obs_last_act'],
},

'planner_unexpected':{
'prompt':"""Did the player encounter any obstacles or unexpected scenarios? Also, did player status change unexpectedly?
Write \"NA\" and end answer here if no expectancy.
Otherwise, should the player modify the subgoals to react?""",
'dep':['s-obs', 's-vit', 'obs_chg', 'obs_last_act'],
},

'planner_mistake':{
'prompt':"""Did the player make any mistakes approaching the top subgoal '$db.subgoals.subgoal$'? Write \"NA\" and stop the answer
↪  here if there's no mistake.
Otherwise, identify the potential causes of the mistake and find the most probable cause by analyzing the current situation.
```

```
Then, suggest a precise and executable modification to the subgoals to address the mistake.""",
'dep':['s-obs', 's-vit', 'obs_chg', 'obs_current_actions'],
},

'challenge':{
'prompt':"""Identify three high level challenges.
Based on the current situation, score their urgency out of 5 (5 is the highest urgency) and score their convenience out of 5 (5
↪  being most convenient).
Then choose a challenge to address based on the urgency, convenience, and overall objective of the game.
Be concise.""",
'dep':['reflect', 'planner_mistake', 'obs_obj', 'obs_inv', 'obs_vit'],
},

'gate-plan_sketch':{
'prompt':"""Reason with the instruction manual and knowledge base. Explain concisely how the choosen challenge may be approached
↪  given the player's current situation, with actions permitted by the game.
Then,
1. Confirm if the subgoal '$db.subgoals.subgoal$' is still accurcate for the choosen challenge.
2. Confirm if the subgoal '$db.subgoals.subgoal$' is incomplete and up-to-date according to the completion criteria
↪  '$db.subgoals.completion_criteria$'.

If yes to both, end the answer here.
Then, sketch an updated high-level plan for addressing the choosen challenge. Do not plan to go back to something if you cannot
↪  provide it's location.
If situation allows, the plan-sketch should aim for achieving relevant unaccomplished achievements in the order of difficulty,
↪  without impacting player safety.""",
'dep':['reflect', 'planner_unexpected', 'planner_mistake', 'obs_obj', 'obs_inv', 'obs_vit', 'achievements', 'challenge'],
},

'gate':{
'prompt':"""Reason with the previous conversation and context, and output a Json dictionary with answers to the following questions:
```
{
"unexpected_encounters": $ANSWER, # [yes/no] Did the player encounter any unexpected scenarios or obstacles?
"mistake": $ANSWER, # [yes/no] Did the player make any mistakes?
"correction_planned": $ANSWER, # [yes/no] Was correction planned at current step for the mistake?
"confused": $ANSWER, # [yes/no] Does the player seem confused?
"top_subgoal_completed": $ANSWER, # [yes/no] Is the most recent top subgoal complete according to the completion criteria?
"top_subgoal_changed": $ANSWER, # [yes/no] Has the most recent top subgoal been changed?
"replan": $ANSWER # [yes/no] Was there a re-plan/change for the plan sketch?
}
```
""",
'dep':['reflect', 'planner_unexpected', 'planner_mistake', 'gate-plan_sketch'],
},

'subgoals':{
'prompt':"""List the top 3 subgoals for the player according to the plan sketch. The subgoals should be quantifiable and actionable
↪  with in-game actions. Put subgoals of highest priority first.
For each subgoal, specify how the completion may be precisely quantified in at most one sentence (include the numbers and details).
Do not include additional information other than the subgoals and quantification.""",
'dep':['reflect', 'obs_obj', 'obs_inv', 'obs_vit', 'gate-plan_sketch'],
},

'top-subgoal':{
'prompt':"""Write the highest priority subgoal and completion criteria as a Json dictionary of the following format:
```
{
"subgoal": $subgoal, # The highest priority subgoal
"completion_criteria": $completion_criteria # The completion criteria for the subgoal
"guide": $guide # A brief high-level guide for completing the subgoal
}
```
""",
'dep':['reflect', 's-action', 'obs_current_actions', 'subgoals'],
},

'subgoal_analysis':{
'prompt':"""Check the instruction manual for requirements to complete the top subgoal.
Identify a list of unknown/missing information/details and do not make any assumptions.
Pay close attention to the details such as the numbers and specifications.""",
'dep':['reflect', 'obs_current_actions', 'top-subgoal'],
},

'skill':{
'prompt':"""First, try to find an existing skill from the 'skill library' that could be used to represent the current top subgoal.

If none of the existing skills could be applied, create a new skill.

The skill, parameters, and guide should be general enough to be reusable for tasks of the same class.

Format the choosen skill as a Json object of the following format:
```
{$skill_name: [$1_line_skill_desciption, $supported_parameters, $skill_guide]}
```
""",
'dep':['top-subgoal'],
},

'planner-adaptive':{
'prompt':"""List and answer at most 3 questions to help guide the agent on the subgoal: '$db.subgoals.subgoal$'. The questions
↪  should prompt the player to think concretely and precisely.
```

```
Example questions could ask the player to:
 - identify goal-relevant details from the observation, history, and instruction manual
 - recall related historical encounters
 - potential obstacles and how to overcome them
 - specify concrete ways to improve efficiency

Output only the list of questions and nothing else. Write \"NA\" if there's no question to ask.""",
'dep':['reflect', 'planner_unexpected', 'planner_mistake', 'top-subgoal', 'subgoal_analysis'],
},

'kb-add':{
'prompt':"""Rewrite the unknown information and details list into a Json dictionary.
Give each item a concise but precise name as the key, and a dictionary of answers to the following inquiries as the value:
```
"item_name":{
"discovered": $ANSWER, # Can this unknown information be precisely uncovered or determined at the current gamestep? [yes/no]
"discovery": $ANSWER, # In concise but precise terms, write what has been uncovered [If the information is not uncovered, write
↪   'NA'.]
"discovery_short": $ANSWER, # Condensed version of 'discovery', only containing precisely the discovered info [If the information
↪   is not uncovered, write "NA".]
"general": $ANSWER, # Confirm that this uncovered information remain unchanged in subsequent steps of this game. [yes/no]
"unknown": $ANSWER, # Confirm that this uncovered information is missing from instruction manual. [yes/no]
"concrete_and_precise": $ANSWER, # Is the uncovered information concrete and precise enough to add to instruction manual? [yes/no]
"solid": $ANSWER, # Confirm that this information is not speculative or deduced based on assumption. [yes/no]
}
```

Include answers with a one-sentence justification or detail. Do not repeat the inquiries.
Write '{}' if there is nothing in the list of 'unknown information and details'.""",
'dep':['planner_unexpected', 'planner_mistake', 'top-subgoal', 'subgoal_analysis', 'obs_obj', 'obs_inv', 'obs_vit', 'obs_chg',
↪   'obs_last_act'],
},

'unknown':{
'prompt':"""Merge the 'unknown/missing information' in the previous answer and the 'Previous unknown information and details' into
↪   a single Json dictionary.
Give each item a concise but precise name as the key, and a dictionary of answers to the following inquiries as the value:
```
"item_name": { # if applicable, use the same item name as in the previous answer or knowledge base
"info": $ANSWER, # In concise but precise language, describe what exactly is missing. [If the information is not missing, write
↪   'NA'.]
"knowledge": $ANSWER, # What do you already know about the current requested info? [If nothing is known, write 'NA'.]
"unknown": $ANSWER, # Confirm that this requested info is missing from the instruction manual. [yes/no]
"novel": $ANSWER, # Confirm that the knowledge base does not already contain precise answer to this requested info. [yes/no]
"general": $ANSWER, # Is the requested info expected to remain unchanged in future game steps? [yes/no]
"relevant": $ANSWER, # Is this requested info helpful for succeeding the game? [yes/no]
"correct": $ANSWER # Confirm that this request does not disagree with the instruction manual. [yes/no] followed by a one-sentence
↪   justification.
}
```
Only include the answers, not the inquires. Remove duplicates and arrange multi-target inquiries into separate items.""",
'dep':['subgoal_analysis'],
},

'actor-reflect':{
'prompt':"""First, identify up to 2 types of information other than observation and action (from the gameplay history) that may be
↪   important for the subgoal '$db.subgoals.subgoal$', and output them in a concise list.

Secondly, summarize the gameplay history using a table.
Use 'TBD' for the action corresponding to the current step, and include the observations, action (including number of steps and
↪   result), and the identified information in the table.

Finally, analyze and evaluate each action in '$db.allowed_actions$'. Utilize spatial and temporal reasoning with the table to
↪   determine if the action is safe, if it leads to an unexplored state.
Format the analysis as a Json dictionary of the following format, and write "NA" for not applicable fields:
```
{
"$action": { # The action name
"spatial_relation_with_current_obs": $answer, # How does this action spatially relate to objects in the observation? Focus on
↪   important objects (1 sentence)
"alignment_with_history":{
"temporal": $answer, # How does this action relate with the historical trajectory temporally? Explain your reasoning by connecting
↪   with the history table. (1 sentence)
"spatial": $answer, # How does this action relate to the historical path spatially? Explain your reasoning by connecting with the
↪   history table. (1 sentence)
},
"risk": $answer, # list potential risks or hazards associated with the action as concisely as possible
"benefit": $answer, # list potential benefits or advantages associated with the action as concisely as possible
},
...
}
""",
'dep':['top-subgoal', 'subgoal_analysis', 'obs_current_actions', 'obs_obj', 'obs_inv', 'obs_vit'],
},

'actor-plan-sketch':{
'prompt':"""First, examine the current observation/surroundings in with a focus on things related to the subgoal
↪   '$db.subgoals.subgoal$' and answer the following questions (no more than 1 sentence per-answer):
Out of the goal-relevant objects, which ones are easily reachable and which ones are harder to reach?
Are there direct obstacles in the way?
Are there risks in the way? Are they addressable?
```

```
Then, determine if the previous plan '$db.action_summary.plan-sketch$' still applies for the subgoal based on the criteiras and the
↪ expiration condition.
Relevance criteria: $db.action_summary.relevance-crieria$
Expiration condition: $db.action_summary.expiration-condition$

If the plan still applies, examine the current observation for any obstacles, hazards, or unexpected scenarios and reason
↪ spatially how they may be addressed.
If necessary, update only the 'details' of the previous plan to achieve the target '$db.action_summary.target$' of the plan.

If the plan does not apply, explain your reasoning, and write an new plan-sketch.
Reason spatially and temporally with the current observation, the gameplay history, and the action analysis.

Finally output the updated plan-sketch. The plan-sketch details should concretely describe a procedure or a specific direction to
↪ follow, and must be a Json dictionary of the following format:
```
{
"plan-sketch": $plan-sketch, # A 1-line summary of the plan-sketch
"detials", # Concrete description of what procedure or a specific direction to follow. Do not offer multiple options or
↪ possibilities.
"target", # Concisely write the target of the plan
"relevance-crieria": $relevance_criteria, # The criteria that the plan is relevant to the current situation
"expiration-condition": $expiration_condition, # The condition that may cause the plan to expire, like specific changes in the
↪ observation or the inventory, or after a certain number of steps.
"notes": $notes # Anything about the current situation or reasoning that may be meaningful to remember for the future.
}
```
""",
'dep':['planner_unexpected', 'planner_mistake', 'top-subgoal', 'subgoal_analysis', 'obs_obj', 'obs_inv', 'obs_vit', 's-action',
↪ 'obs_current_actions', 'actor-reflect'],
},

'actor-actions':{
'prompt':"""Given the target: $db.action_summary.target$

First, identify any hazards in the observation/surrounding, and identify any obstacles that may interfere with achieving the target.
- Plan-sketch: $db.action_summary.plan-sketch$
- Plan details: $db.action_summary.details$

Discuss how to spatially address or evade the hazards and obstacles based on analysis of the observation, target, and the
↪ plan-sketch.

Then, reason with the instruction manual and the current observation, and identify a sequence of actions to achieve the target.

The sequence of identified actions should start from current step and only include actions for up to the first 6 steps, without
↪ skipping any steps.
Think step by step and explain your reasoning before arriving at the answer.

Group repeated actions together to speed up the game, by explicitly stating the number of repeats for each action.

** Note **:
- The player's reach is 1 step to the front.
- If an object is k steps to a *straight line in the facing direction*, you only need *k-1 steps* to reach the object.
""",
'dep':['obs_inv', 'obs_vit', 's-action', 'obs_current_actions', 'obs_obj'],
},

'actor-final':{
'prompt':"""Examine the actor plan and reasoning and identify the first action (only the exact action name from the list of all
↪ actions in the manual) in the plan.
Then, if the identified action is *explicitly stated as a repeat*, write the number of times this action should be repeated.
Finally, identify if there are any observed/nearby hazards or obstacles, no matter if they interfere with the plan.
Format the output as a Json dictionary of the following format:
```
{
"action": $action, # Write only the exact action name from the list of all actions in the manual.
"repeats": $repeats # The number of times this action should be repeated, write 1 if the action is not stated as repeated.
"hazard": $hazard # [yes/no] Presence of hazards or obstacles in the observation/surroundings, no matter if they interfere with the
↪ plan.
}
```
""",
'dep':['obs_obj', 'actor-actions'],
},

's-plan':{
'prompt':"In one sentence, describe the high-level subgoals and the reasoning behind the subgoals.",
'dep':['reflect', 'planner_unexpected', 'planner_mistake', 'top-subgoal', 'subgoal_analysis', 'actor-reflect'],
},

}
```

## F.2 Webshop Prompts

```
{

'obs': {
'prompt':"""Describe the products and interactible elements on the page as Json dictionaries:
products =
{
```

```
...
}
interactible_elements =
{
...
}
""",
'dep': [],
},

'task_filter':{
'prompt':"""If the webpage does not contain any products, output 'N/A' and end the answer here.

Otherwise, examine the current webpage and the task:
```
$db.environment.refined_task$
```

and output a Json dictionary explaining how the product(s) on the page match the task specifications.Check the guidance for criterias to match the products.
The Json dictionary should contain for each product:
 - brief reasoning
 - estimated match-percentage (equally weight for all features)
 - estimated match-percentage *without taking into account of any mismatch in customizable features* (equally weight for all, but not including customizable)
 - estimated importance-adjusted match-percentage (weighting the features by importance)
 - overall match-percentage (the highest of the above three)
""",
'dep': ['obs'],
}

'actor_sketch':{
'prompt':"""First, identify up to 2 types of information other than webpage and action (from the browsing history) that may be important for the task,
and output them in a concise list.

Secondly, summarize the websites, actions, and the identified information from the browsing history using a table.
Use 'TBD' for the action corresponding to the current step.
Be precise with the chronological order of each step.

Third, describe the current webpage in less than two sentences, drawing connection to the task:
```
$db.environment.refined_task$
```

Then, reason with the browsing summary, the current webpage, and the guidance to determine how the task may be addressed with actions permitted by the website.
Explain your reasoning before arriving at the answer.

Then, determine if the plan from the most recent step should be continued.
Finally, draw an updated plan-sketch starting from the current step, only consisting of actions permitted by the website.""",
'dep': ['obs', 'task_filter'],
},

'action':{
'prompt': """Using information from the plan-sketch, write an actionable plan to approach the task:
```
$db.environment.refined_task$
```

Then identify the best action to take at the current step, and output the action as a Json dictionary of the following format:
```
{
"action": $action, # search/click
"target": $target, # the target of the action
}
```
""",
'dep': ['actor_sketch'],
}

'summary_obs': {
'prompt':"""In one sentence, how the items on the page match the task.
```
$db.environment.refined_task$
```
Only provide details on the item that match the task the most (when measurements are present, provide the highest).
""",
'dep': ['task_filter'],
},

'summary_actor_plan': {
'prompt':"""In one sentence, describe the plan in detail, including future actions. Then in one sentence, describe the reasoning behind the plan.""",
'dep': ['actor_sketch', 'action'],
},
}
```

# G  UI

```
DAG Manager Dashboard
---------------------

Current DAG Nodes and their details:
The graph is currently empty.

Options:
 [A] Add node
 [E] Edit node
 [L] Load graph
 [S] Save graph
 [R] Evaluate graph
 [X] Exit

Choose an action (A, E, L, S, R, X): █
```

Figure 5: Screenshot of our command line interface (CLI) to produce a graph without coding.

