# OpenReview forum: "AgentKit: Structured LLM Reasoning with Dynamic Graphs"
_colmweb.org/COLM/2024/Conference — COLM_

### Official Review · Reviewer_geit · 2024-05-08

**Rating:** 8
**Confidence:** 4
**Ethics Flag:** 1

**Summary:**

This paper presents a novel framework for structuring the reasoning process of large language models to tackle complex tasks. The use of DAG for structuring the thought process is technically sound. The authors provide quantitative results demonstrating the effectiveness of their framework, showing SOTA performance on WebShop and Crafter. It also gives a detailed explanation of the AgentKit framework and its organization aids in understanding the flow of ideas and the progression of the research.The framework has the potential to impact how LLMs are used in a variety of applications. Overall, it presents a high-quality, clear and significant work.

**Reasons To Accept:**

This paper presents a framework for natural language “coding” of end-to-end multifunctional AI agents, with the abilities to plan, reflect, and learn under challenging situations. This approach is innovative in its use of a node-based system to simulate a structured thought process. And it could be useful in many real-world tasks. Also the authors will open source its GitHub repository, which can lead to reproducibility and further community involvement.

**Reasons To Reject:**

The paper may lack a thorough comparative analysis with similar frameworks.

---

> ### Author Rebuttal · Authors · 2024-05-26
>
> Thank you for the recognition that our framework is **novel** and has the potential of **impacting** how LLMs are used in **real-world tasks**, and thank you for the recognition that our work is of **high-quality**.
>
> # Evaluation
> We include comparison with Act, React, and the custom framework by AgentBench in WebShop. Please note that since frameworks like React/Act requires few-shot demonstrations, we are unable to apply them to Crafter due to the trajectory length (can go easily above 200 steps).
>
> # Changes
> Updated the repo github.com/anonymousLLM/AgentKit with replacement to Figure 1 to better showcase contribution.

---

> > ### Comment · Reviewer_geit · 2024-06-06
> >
> > OK. Thanks for your response.

---

### Official Review · Reviewer_GbVx · 2024-05-10

**Rating:** 4
**Confidence:** 4
**Ethics Flag:** 1

**Summary:**

This paper proposes a framework (AgentKit) that constructs a graph with LLM prompts for reasoning, where each node contains a natural language prompt for a specific subtask. The paper claims that the proposed framework has modular interpretability and is easy to use (without the need to write explicit code). Experiments on Crafter and Webshop shows the framework works well and can generalize to different environments.

**Reasons To Accept:**

1. The proposed framework is general, and intuitively makes sense, I would expect it to work on a diverse set of tasks.
2. The idea of incorporating dynamic graph into LLMs for reasoning could have high potential.

**Reasons To Reject:**

1. Despite the high potential of the graph structure, it's unclear from the experiments whether it actually helps. First, the performance is (significantly) worse than Spring (20.6 vs 27.3) on Crafter. And there's no clear explanation from the paper why it is worse; Second, on Webshop, the improvement is also unclear. Actually, I'm not sure if the reported baseline (ReAct) uses GPT-3 or GPT-4 as base LLM, and the paper does not mention it as well. Therefore, I think the results of the proposed approach are quite weak to validate its effectiveness.
2. The writing of the paper needs significant improvement. First, it's very hard to get where the key motivation is. The paper proposes multiple components, but it's not explained well why we need this component. e.g., knowledge base (do we actually need an explicit KB, or just list all possible actions?). Second, some of the phrases are not accurate. E.g., in conclusion, it says "State-of-the-art performance" which is not accurate, as on Crafter, the performance is significantly worse.

To summarize, the proposed framework is a bit overcomplicated, while the effectiveness over simpler baselines is not validated; Also the authors need to make the writing clear, especially in introducing the key motivations and the intuitions of different components.

---

> ### Author Rebuttal · Authors · 2024-05-26
>
> Our core contribution is 1.SOTA agents in Crafter and WebShop from a unified structure of “nodes”, 2. a **unified** framework that enables for developing **multi-functional** agents, with SOTA performance in multiple tasks.
>
> ## 1.1 Crafter
> Apologies for the confusion. Details on the score vs. reward for SPRING and AgentKit are in Sec. 4.1.1, 4.1.2
> - AgentKit outperforms SPRING with GPT-4-turbo by 80% and costs 45% less than SPRING with GPT-4, while achieving slightly higher average (SOTA) rewards
> - Score favors random behaviors (unlocking more achievements at lower probabilities)
>
> SPRING relies on randomness to get past stuck situations (Section 4.1.2), whereas AgentKit uses *planning and learning from mistakes*. SPRING tends to unlock less important achievements (e.g., Eat Plant, Collect Sampling), **improving the score but not benefiting survival**. Therefore, using score to determine the inferiority of AgentKit is **unfair**, as the score metric favors RL agents with better exploration.
>
> ## 1.2 WebShop
> Thank you for pointing out the omission. We will update the final version to include the LLMs used by baselines in the WebShop table. Specifically, Act and React use PaLM-540B, and AgentBench reports the best of LLMs (including GPT-3.5/4). Note that AgentBench and ReAct are **few-shot**, whereas AgentKit is **zero-shot**, making the task harder for AgentKit. Please take this into consideration when evaluating the fairness of the comparison.
>
> ## 2.1 Components
> Thank you for highlighting the confusion. We will update the presentation to better state our core contribution. The primary contribution of AgentKit is an easy-to-use framework for assembling simple language subtasks for multifunctional agents. **Multiple** important agent features work together under a **unified and modular framework**. Our experiments show generalization to different and challenging domains: game and web in a **zero-shot** manner.
>
> ## 2.2 Complexity
> We include qualitative analysis and examples of Planning, Reflection, and Learning in Section 4.1.2 and Figure 4. AgentKit's design simplifies building agents to writing a list of natural language subtasks and dependencies. While we showcase a Crafter agent with sophisticated capabilities, the underlying building blocks are **unified** in **natural language form** (Appendix E) and transfer easily to the web.
>
> ## Changes
> Updated github.com/anonymousLLM/AgentKit with a replacement for Figure 1 to better showcase our contribution.

---

> ### Author Response · Authors · 2024-06-03
>
> Dear reviewer,
>
>    Thank you again for your review. Please let us know if we are able to address all your concerns.

---

### Official Review · Reviewer_Lfw9 · 2024-05-11

**Rating:** 6
**Confidence:** 2
**Ethics Flag:** 1

**Summary:**

The paper introduces AgentKit, a prompting framework for multifunctional agents. AgentKit uses simple natural language prompts to explicitly build complex "thought processes." The framework's core component is a node, which includes a prompt for a specific subtask. Users can combine these nodes to form a structured "thought process". Quantitative results show that agents designed using AgentKit achieve SOTA performance on Webshop and Crafter.

**Reasons To Accept:**

**Comprehensive, reliable and detailed agent framework**: AgentKit is a comprehensive agent framework that incorporates numerous essential functionalities for critical reasoning tasks. The framework offers transparent reasoning steps, enabling users to easily identify and rectify any reasoning errors. This makes it particularly valuable for those who aim to diagnose and improve their reasoning processes.

**Reasons To Reject:**

**1. Novelty of the framework**: In my view, this framework resembles an advanced self-reflection agent system, enhanced by specialized knowledge modules. Several unique nodes are employed to summarize previous trajectories and guide planning at the current step. However, this also brings to mind two relevant works: CLIN (Majumder et al., 2023) and SSO (Nottingham et al., 2024). Both incorporate memory modules (akin to a database) that refine agent skills. Thus, while this framework is quite sophisticated, it fundamentally builds on prior research with incremental, finer-grained modifications.

- Majumder et al., 2023. CLIN: A Continually Learning Language Agent for Rapid Task Adaptation and Generalization.
- Nottingham et al., 2024. Skill Set Optimization:Reinforcing Language Model Behavior via Transferable Skills.

**2. Lack of in-depth evaluation**: The authors present only two performance tables for two tasks without a more detailed, fine-grained analysis, leaving several critical questions unanswered. For instance, what impact would the removal of node types n_{kg add} and n_{unk} have on performance? If the context window length were reduced from 25 to 20, would there be a significant performance decline? What role does the plan summarization node play in the overall performance, and how crucial is dynamic editing for the graph structure? Furthermore, given the framework's complexity, providing qualitative examples, particularly for the self-driving task, would be essential for helping readers understand the workflow and better grasp the practical implications.

---

> ### Author Rebuttal · Authors · 2024-05-26
>
> Thank you for recognizing that AgentKit is comprehensive, reliable, and detailed, which can serve users in diagnosing and improving LLM agents for a wide range of tasks.
>
> # 1 Novel
> We will update our final paper to cite Majumder et al., 2023, and Nottingham et al., 2024. These works focus on extracting `knowledge` or `prompts` with a curriculum involving multiple turns with the environment. Key differences in our setting include:
> 1. **Contribution** The main contribution of AgentKit is a unified, easy-to-use framework for **multi-functional** agents, whereas existing works focus on one functionality. AgentKit can be combined with Nottingham et al. for learning from multi-turn interactions.
> 2. **Use of reward/feedback**: AgentKit never uses environment rewards.
> 3. **Rollout length**: Crafter spans 150-300 steps, up to 500, making credit assignment challenging and token-costly.
> 4. **Subtask length**: Simple subtasks in Crafter (e.g., `find_wood`) can take over 10 steps; complex ones (e.g., finding a diamond) over 100 steps. Nottingham et al. tested with subtask length <=5, scaling might be an issue.
> 5. **Environment noise**: Crafter has conflicting objectives (e.g., survival vs. tech progress) and distractions from procedural generation and random spawning, which present challenge as noted by Nottingham et al.
>
> Note: Both papers are recent, with one accepted by ICML2024 and the other released less than 2 months before the COLM deadline, making it difficult to build AgentKit on top.
>
> # 2.1 Eval.
> AgentKit supports **multifunctional** agents, unlike individual features like planning, reflection, and KB. Our Crafter agent demonstrates three advanced agent functionalities **working together** under AgentKit's unified framework. Our experiments highlight generalization to **distinct** and challenging domains (a game and the web) in a **zero-shot** setting.
>
> # 2.2 Example & Detail
> Qualitative analysis and examples of Planning, Reflection, and Learning are in section 4.1.2 and Figure 4.
> - Remove KB: Slower progress past stuck situations (sec 4.1.2), leading to worse performance.
> - Plan summarization: Necessary due to lengthy and noisy observations.
> - Context window: [20,30] range should be fine, depending on the game.
> - Dynamic graph: Reduces cost by 1/3 with no performance impact.
>
> # Changes
> - We will add detailed information in the final version.
> - Updated the repo github.com/anonymousLLM/AgentKit with a replacement for Figure 1 to showcase contributions.

---

> ### Author Response · Authors · 2024-06-03
>
> Dear reviewer Lfw9,
>
>    Thank you again for your review. Please let us know if we are able to address all your concerns.

---

> > ### Comment · Reviewer_Lfw9 · 2024-06-04
> >
> > Thanks for the responses!
> >
> > My concerns have been partially addressed, but I still seek clarification on the term "multifunctional." Can the simple ReAct agent be considered multifunctional because it can plan, ground actions, and access KB? While the proposed framework appears advanced, it is challenging to comprehend the benefits the whole framework and part of it offers. I find the authors' explanation in the rebuttal is not convincing, as they merely provide a brief description without supporting figures or detailed elaborations. Furthermore, there seem to be many parts within the framework that require further investigation. As a result, I think the work as somewhat incomplete, and I would appreciate additional evidence and clarifications in the rebuttal and future iterations. Therefore, I maintain my score for now.

---

> > > ### Author Response · Authors · 2024-06-04
> > >
> > > Dear reviewer,
> > >
> > > By multifunctional, we mean explicitly multifunctional. React can be prompted with multifunctional examples, and hopefully the LLM will do whatever intended. However, there is no explicit control over the outputs and this problem presents even more challenge in hard tasks. ReAct demonstrates poor performance on demanding tasks, also noted by [1].
> > >
> > > We include results on 2 very challenging tasks: crafter and Webshop. As shown in Table 2, AgentKit outperforms ReAct and AgentBench (which uses a similar prompt to react/reflexion).
> > >
> > >
> > > [1] Voyager: An Open-Ended Embodied Agent with Large Language Models
> > >
> > >
> > > Finally, we note that our work is already at the page limit of this conference. We provide an easy-to-use and well-documented codebase, we leave further investigations to future works.

---

> > > > ### Comment · Reviewer_Lfw9 · 2024-06-04
> > > >
> > > > Thanks for the quick response!
> > > >
> > > > The explanation has clarified the concept to some extent, but I still find it very challenging to grasp the effect of the modules within the framework. Given the complexity of this method, it's essential to clearly outline the motivation for incorporating specific modules and support this with comprehensive experiments that demonstrate their necessity. Unfortunately, this explanation does not achieve that clarity. This isn't an issue of page limits, as additional details could always be included in the appendix. I maintain my current score.

---

> > > > ### Author Response · Authors · 2024-06-04
> > > >
> > > > We ask that the reviewer consider instead the contribution of AgentKit as an easy-to-use toolkit for controlling LLM behavior through “thought-process”.
> > > >
> > > > The framework itself is **very simple**: just graphs and nodes(prompts). See https://github.com/anonymousLLM/AgentKit
> > > >
> > > > Although we showcase some complicated multifunctional agents with competitive performance on Crafter, AgentKit could be used to **build simple agents**. For example, the agent we built for Webshop only consists of a planner (appendix E.2).

---

> > > > > ### Comment · Reviewer_Lfw9 · 2024-06-04
> > > > >
> > > > > I greatly appreciate the authors' efforts in developing an easy-to-use tool that works across various agent task types. However, ease of use from an engineering perspective does not necessarily imply that the effects of the modules and their comparison with other possible variants have been clearly explained. This clarity is crucial for a research paper. Overall, the paper is impressive, but it still lacks the fine-grained analysis needed to convincingly demonstrate that this is one of the optimal frameworks.

---

> > > > > > ### Author Response · Authors · 2024-06-04
> > > > > >
> > > > > > Dear Reviewer,
> > > > > >
> > > > > >   Thanks for the quick response and continued engagement, and for raising this important concern. We hope to stress that our goal is to present AgentKit as a simple and general framework for **prompting** LLMs on agentic tasks. (AgentKit also demonstrates  a new perspective for creating LLM agents)
> > > > > >
> > > > > >   To show **generalization**, we demonstrate SOTA performance through fair comparison in two very different and challenging tasks.
> > > > > >   To show **simplicity**, we offer a screenshot of the UI we developed to build agents without a single line of coding (in GitHub repo).
> > > > > >   To show **effectiveness** we compare to React, Act, and AgentBench(react/reflection style) and show superior performance on Webshop.
> > > > > >
> > > > > >   We understand that the Crafter agent contains a lot of complicated features, we hope that the reviewer consider the contributions listed above independent from the hyper-parameter/module details of Crafter (since all the prompts have been released).

---

> > > > > > > ### Comment · Reviewer_Lfw9 · 2024-06-06
> > > > > > >
> > > > > > > Thank you for the great response! I will raise my score, but I still hope that the authors could provide more fine-grained analysis about why the module design could produce better performance to fully convince the readers.

---

> > > > > > > > ### Author Response · Authors · 2024-06-06
> > > > > > > >
> > > > > > > > Dear reviewer,
> > > > > > > >  Thank you for actively engaging in the conversation, providing the insightful feedback, and raising your score in support of our contribution.
> > > > > > > >  We will address the omitted citations and include more experimental details in the final version about the modules for Crafter.

---

### Official Review · Reviewer_DxSd · 2024-05-12

**Rating:** 6
**Confidence:** 4
**Ethics Flag:** 1

**Summary:**

This paper proposes an abstraction for structured reasoning using LLMs based on a Direct Acyclic Graph (DAG) structure where

1. nodes handle a specific subtask correspond to LLM calls with pre and post-processing
2. directed edges indicate which node outputs become which node inputs

In addition to this static and local processing flow

3. there is a shared database of values that is read and written by every node that allows global communication
4. there is the possibility of dynamically adding or removing edges

the experimental setup features comparison with other agent frameworks on the Crafter and Webshop tasks

**Questions To Authors:**

[S2.3]
>Since there may exist more than one topological order for a given graph, the dynamic addition/removal of components may result in non-deterministic or unexpected behaviors. Safeguards are put in place to catch potential unexpected behaviors.

Could you clarify this statement. If the graph is statically defined and topo-sorted before starting the processing but then dynamic graph changes are allowed, this can cause infinite loops. What are these safeguards?

**Reasons To Accept:**

**[A]** The topic is very timely and explores an area of growing importance: Abstractions needed in building agent systems.

**[B]** The paper proposes a framework with a specific set of abstractions and an experimental setup with two commonly used agent benchmarks (Crafter, Webshop) where to test a particular instantiation of these against existing systems. The setups make sense and the comparison is against recent state of the art systems.

**Reasons To Reject:**

It is hard to judge the contributions to the paper because:

**[A] It is hard to know which components contribute to what**

The paper proposes a framework with a specific set of abstractions but it remains unclear how these help because the have conflicting properties i.e.

1.  the system starts with desirable properties that bring simplicity and understandability. These are

    a. locality: provided by imposing a DAG structure. This indeed allows to think of sub-tasks independently and diagnose better
    b. static behavior: provided by having this DAG being pre-defined before execution

2. but then it adds properties that have the opposite effect, to attain flexibility and generality

    a. globally shared variables through a data-base
    b. dynamic modification of the DAG via "an API"

so its hard to know if the locality/staticity help, what are the tradeoffs with regard to flexibility and generality. The paper just presents aggregate results with no ablations. For all we know the conflicting properties above give enough freedom to write any "neural computer program" and would make this paper more of a software package technical report, with not a strong research component.

A possible improvement would be to compare local/static implementations versus the full version to analyze the effect of these design choices.

**[B] The experimental setup for the Crafter setup has important confounding factors**

It is well known that agent systems are very dependent on the LLM used, e.g. switching from GPT-4 to GTP-3.5 severely affecting performance [(Wang et al 2023)](https://arxiv.org/abs/2305.16291)

But the experimental setup compares system with multiple different LLMs including some with markedly different performance

>Our AgentKit implementation uses a combination of GPT-4-0613, GPT-4-turbo, and GPT-3.5- turbo to balance cost and performance

This is not the authors fault. Its a sad state of affairs that closed-source proprietary systems are so far ahead open source alternatives for agent systems, but this does not change the fact that it is hard to judge results (i.e. was it the use of GPT-4 at some task that made the different?, have the authors identified tasks or flows that for which GPT-3.5 is good enough and competitive with GPT-4. All of this would be important for the papers scientific contribution.)

It should be noted that the experimental setup for Webshop **does** have comparable setup using GPT-4 but also that it only uses the first 100 samples of Webshop, a specific category of Agent Bench, see

[Table 3 in (Liu et al; 2023)](https://arxiv.org/abs/2308.03688)

Agent Bench contains many more frameworks that were left out. So the setup for this second task could also be stronger. Again this is not exactly the authors fault.

---

> ### Author Rebuttal · Authors · 2024-05-26
>
> Thank you for recognizing that our paper explores a timely topic and our setup/evaluation make sense.
>
> # [A] local vs. global
> Please note a fully static process has been explored by [1]. We believe that **global components (e.g., memory, planning) are indispensable to AI agents**.
>
> The primary contribution of AgentKit is an easy-to-use **framework** to build agents with **multiple** global features **working together**, with modular, simple-to-understand subtasks and DB. Our experiments focus on showing generalization to **different** and challenging domains: game and web, in a **zero-shot** manner.
> For more insights on static vs. multifunctional implementations, consider SPRING [1] vs. AgentKit in S4.1.1. AgentKit quantitatively demonstrates reduced cost and higher consistency and qualitatively demonstrates more advanced capabilities (S4.1.2).
>
> Finally, note that `branching` only is used in a user-controlled setting: skipping planner when there's no plan updates.
>
> ## Ablations
> As shown in appendix E, the `static/local` prompts are designed to achieve global features by using a database. Removing the global functions plan, reflect, KB will result in a completely different design.
>
> # [B] Mixing models
> Please note that the baseline on Crafter [1] also uses GPT-4 for fair comparison. Insights on mixing GPTs: `GPT-4` dominates `3.5` in performance. We use GPT-3.5 for `s-X` and some `obs-X` (summarization/rewrite). We use `4-turbo` for the rest instead of `0613` except `actor-reflect` because `4-turbo` has trouble reasoning with space (sometimes cannot tell north opposite to south).
>
> ## WebShop
> Note that both AgentBench and ReAct are **few-shot** and AgentKit is **zero-shot**, which makes the task harder for AgentKit.
>
> [1] SPRING: Studying Papers and Reasoning to play Games
>
> # Q [S2.3]
> For dynamic add/remove of edge (u,v), AgentKit asserts: `v` have not been evaluated for `add` and both `u,v` have not been evaluated for `remove`. The assert statements avoid not only loops but also unpredictable behaviors. Consider a graph `A-F, C-F` with multiple topological orders. A dynamic edge `C-A` could potentially alter the previous topological order. The assert statements raise it to the attention of the user. A loop can still be implemented by adding new nodes (chain A-B-… that keeps growing) and it is up to user to prevent infinite loops.
>
> # Changes
> Updated the repo github.com/anonymousLLM/AgentKit with replacement to Figure 1 to better showcase contribution.

---

> > ### Comment · Reviewer_DxSd · 2024-05-31
> >
> > I would like to thank the authors for the explanations I understand better now the aspect of dynamic edge editing constraints. No changes for my views about the paper overall.

---

### Decision · Program_Chairs · 2024-07-10

**Decision:**

Accept

**Comment:**

The paper introduces a rather complicated graph (DAG)-based framework to learn agent behaviors and shows positive results on benchmarks such as Webshop and Crafter. The majority of reviewers agree that this paper has enough merits to warrant acceptance. There has been good faith discussion during the rebuttal period where the authors have attempted to address reviewers' concerns. However, the paper's writing has certain clarity issues, and some reviewers suggested discussing the strengths of the system and its novelty. We suggest the authors incorporate the reviewers' suggestions in the final version to make the paper more accessible to readers who are not specifically working in this space.

[At least one review was discounted during the decision process due to quality]